# Development and Validation of Novel HPLC Methods for Quantitative Determination of Vitamin D3 in Tablet Dosage Form

**DOI:** 10.3390/ph17040505

**Published:** 2024-04-15

**Authors:** Muhammad Saqib Gohar, Taj Ur Rahman, Ali Bahadur, Ashraf Ali, Sarah Alharthi, Nora Hamad Al-Shaalan

**Affiliations:** 1Department of Chemistry, Mohi-Ud-Din Islamic University, Nerian Sharif, Azad Jammu & Kashmir 12080, Pakistan; saqibgohar020@yahoo.com (M.S.G.); taj_urrehman81@yahoo.co.uk (T.U.R.); 2Department of Chemistry, School of Natural Sciences (SNS), National University of Science and Technology (NUST), H-12, Islamabad 46000, Pakistan; alibahadur138@gmail.com; 3School of Chemistry & Chemical Engineering, Henan University of Technology, Zhengzhou 450001, China; 4Department of Chemistry, College of Science, Taif University, P.O. Box 11099, Taif 21944, Saudi Arabia; sarah.alharthi@tu.edu.sa; 5Research Center of Basic Sciences, Engineering and High Altitude, Taif University, P.O. Box 11099, Taif 21944, Saudi Arabia; 6Department of Chemistry, College of Science, Princess Nourah bint Abdulrahman University, P.O. Box 84428, Riyadh 11671, Saudi Arabia; nhalshaalan@pnu.edu.sa

**Keywords:** vitamin D3, method development, limit of quantification, high-performance liquid chromatography, limit of detection

## Abstract

In the present work, an efficient isocratic HPLC method was developed for the precise and accurate estimation of vitamin D_3_ in tablet form. The chromatographic conditions comprised an L3 silica column (5 µm in particle size, 4.6 mm × 250 mm) with a mobile phase n-hexane/ethyl acetate (85:15 *v*/*v*) with a flow rate of 2.0 mL/min and a detection wavelength of 292 nm. The new methodology was validated for accuracy, precision, specificity, robustness, and quantification limits according to an official monograph of USP/BP and ICH guidelines. The peak areas of the six replicates of the homogeneous sample were recorded. The mean value obtained was 67,301, and the relative standard deviation (RSD) was 0.1741. The linearity and range were in the acceptable bounds, i.e., 0.999, which was calculated using regression line analysis. The results show that the method is truly acceptable as the RSD, as the flow rate was 0.81%, while for the mobile phase composition, it was 0.72%, which lies in the acceptable range. The limit of detection (LOD) and the limit of quantification (LOQ) values were 0.0539 µg/mL and 0.1633 µg/mL, respectively. The % RSD of the intra and inter-day precision of the method was deemed acceptable according to the international commission for harmonization guidelines. The developed method has potential to be used for the detection and quantification of vitamin D_3_ during routine analysis for tablets in dosage form.

## 1. Introduction

The analytical determination of bulk drug materials, intermediates, impurities, drug formulations, degradation products, and related metabolites is of vital importance in pharmaceuticals [1,2]. The methods of analysis for pharmaceuticals are considerably less complex than the analysis of drugs and their metabolites in plasma and biological samples such as blood, urine, or hair [3,4]. Nevertheless, the unambiguous determination of drugs in pharmaceutical formulations is equally important because the quality control of pharmaceutical products is directly related to the health of the patient [5,6]. In pharmaceutical control and drug development, chemical analysis plays a vital role in ensuring high safety and efficacy for patients. Because of this reason, appropriate and authentic methods of quality control are of predominant importance within the pharmaceutical industry [7,8]. Pharmaceutical research and development resulting in the formation of highly complex molecules and drug formulations, and therefore highly selective and novel analytical techniques, are required for their separation and purification [9,10,11]. Thus, appropriate analytical methods should be developed for controlling the quality of pharmaceutical analysis [8,12]. Various techniques like spectrophotometry, electroanalytical techniques (majorly voltammetry), titrimetric, fluorimetry, and chromatographic methods like TLC, GC, HPLC, and CE (capillary electrophoresis) are used for the identification and quantitative analysis of pharmaceutical drugs [1,5,13,14,15].

Vitamin D (cholecalciferol) plays a vital role in the human body [16]. A deficiency of vitamin D causes rickets in children, an exacerbation of osteoporosis, muscle weakness, and abnormalities in the metabolisms of both phosphorus and calcium in humans [17]. Vitamin D is available in different dosage forms, such as tablets, syrups, soft gel capsules, and injectables [18]. A deficiency of vitamins can lead to a suppression in the whole immune system through affecting adaptive and cell-mediated immune responses, which ultimately results in impaired immune response regulation and enhanced malnutrition, morbidity, and mortality [19,20]. Therefore, the intake of essential vitamins and trace elements in the proper amounts in our daily diet plan can result in facilitating the relative powers of the human body, which ultimately results in health and wellbeing [21]. Vitamin D is also critical for ensuring maximum immune functionality, and produces antimicrobial proteins (peptides) like cathelicidin, which combat different pathogens such as mycobacterium tuberculosis in human cells (macrophages) [22,23].

The main issue in the qualitative and quantitative analysis of vitamin D is due to the low concentrations in prescription medicines and nutritional supplements, and therefore requires a precise and robust method for analysis [24,25,26]. HPLC is widely used in pharmaceutical analysis, owing to its non-destructive nature, rapid separation, sample recovery, and robustness [27]. Recently, researchers have reported HPLC- and LC-MS-based methods for the analysis of different analytes, such as vitamins, proteins, saccharides, enzymes, etc. Liquid chromatography coupled with mass spectrometry (LC-MS) has also been used for the analysis of various analytes, including vitamin D in pharmaceuticals, foods, and feeds, in order to attain more information regarding the structure of each analyte from their mass spectra [28,29,30,31]. Due to the non-availability of sophisticated LC-MS systems, chemists and pharmacists are attempting to develop a simple HPLC method for the analysis of targeted analytes. 

Various HPLC-based methods have been developed for the analysis of vitamin D3 in pharmaceutical formulations [32,33,34,35]. The problems associated with the developed methods have a lower stability and a poor selectivity, as well as requiring a longer run time, the use of costly solvents for the preparation of mobile phase and highly complicated sample preparation, such as supercritical fluid or solid-phase extraction, prior to HPLC separation [31,32,35,36,37,38,39]. It is therefore necessary to develop a highly sensitive, selective, less time-consuming, cost-effective, and simple method for the analysis of analytes (Vitamin D3) in pharmaceutical and biological samples [37]. Recently, Jehangir et al. [36] have reported a reverse phase HPLC method for vitamin D3 and menaquinone-7 (MK-7) in tablet form, using gradient elution conditions and methanol/water (95:5 (*v*/*v*)) as mobile phase A, and using isopropyl alcohol/triethylamine (99.5:0.5 *v*/*v*) as mobile phase B. The separation was carried out with ultrahigh-pressure liquid chromatography (UHPLC) using ACE Excel 2 C18-PFP column. Similarly, Themova and Roskar [37] have developed an LC-MS method for the determination of vitamin D3, vitamin E-acetate, K1, A-palmitate, coenzyme Q10, and β-carotene using acetonitrile/water (99:1, *v*/*v*) as the mobile phase. The analytes were identified using LC-MS method with an Agilent Infinity 1290 LC system and an electrospray ionization mass spectrometer (ESI-MS). Although these methods have shown good separation for vitamin D3, the method developed by Jehangir et al. [36] is a gradient separation method in which multiple mobile phase composition changes occur during analysis. There are several limitations of gradient elusion, such as the optimization of the gradient program, selection of suitable solvent composition during analysis, and gradient slopes, which can take a long time and can include a lot of trial and error. Furthermore, gradient elution may require more sophisticated equipment and solvent management techniques. The precise control of the solvent mixing and delivery systems is necessary due to the fluctuating composition of the mobile phase. On the other hand, isocratic separation is simple and less time consuming because the same mobile phase is used throughout the analysis. Therefore, an isocratic separation method was used in our current study to develop a simple method. Similarly, in the second method from Themova and Roskar [37], the separation was carried out using liquid chromatography coupled with mass spectrometry (LC-MS). Again, LC-MS is a sophisticated machine which is not available in every laboratory. In contrary to the above methods, we have developed a simple isocratic HPLC method for the analysis of vitamin D3 in tablet form with excellent separation efficiency. Therefore, our current method is a simple gradient method using a single mobile phase throughout the analysis, and the method is validated and reproducible.

In the current study, a simple normal phase HPLC method was developed for the quantification of vitamin D3 in tablet form. It was found that the developed method was easy, reliable, and eco-friendly for the detection of vitamin D3 in drug form. The developed method was validated for accuracy, precision, specificity, robustness, and quantification limits according to an official monograph of USP/BP and ICH guidelines. The tolerance and authenticity of the method was validated via the application of all international validation protocols. The developed method has a low limit of detection and limit of quantification, and the method is reproducible. Thus, this simple isocratic NP-HPLC method could be used for the detection and quantification of vitamin D3 during the routine analysis of tablet forms in laboratories. 

## 2. Results and Discussion

### 2.1. Method Development 

After several trials of mobile phase selection, n-hexane and ethyl acetate were selected for the normal phase, and acetonitrile, 2-propanol, and methanol were chosen for the reverse phase in various proportions. For the reverse phase, the study was carried out at 275 nm, and for the normal phase, the wavelength selected was 292 nm. A mobile phase consisting of n-hexane and ethyl acetate with a ratio of 85:15 (*v*:*v*) was selected for the separation of vitamin D3. A flow rate of 2.0 mL/min provided appropriate separations with reasonable run times. 

Using a normal-phase silica column (L3), the retention time for vitamin D_3_ was observed to be approximately 4.8 min, and the maximum absorption of vitamin D_3_ was recorded at 292 nm. Hence, the wavelength selected for the analysis was 292 nm. A chromatogram obtained using a normal-phase silica column (L3) and a mobile phase consisting of n-hexane and ethyl acetate with a ratio of 85:15 (*v*:*v*) and a UV detector wavelength of 292 nm is shown in Figure 1.

### 2.2. Method Validation

#### 2.2.1. System Suitability

For system suitability, the peak areas of the six replicates of the homogeneous sample were determined by injecting the samples with the above-stated method, and the following results were obtained as shown in Figure 2. The retention time for vitamin D was approximately 4.8 min. The data of system suitability, i.e., the peak area of various replicates, their standard deviation, and their relative standard deviation, is shown in Appendix A. The peak areas of the six replicates of the homogeneous sample were recorded. The mean value obtained was 67,301; the standard deviation was 117; and the relative standard deviation was 0.1741%. The acceptance criteria included the RSD being <2.0%; the relative standard deviation of the six replicates was less than 2%. Therefore, the system’s suitability for the test was considered acceptable. A chromatogram for system suitability is provided below.

#### 2.2.2. Linearity and Range

For the establishment of the linearity concentration ranges of a test substance, the samples were prepared at concentrations from 80% to 120% and injected, and the chromatograms were recorded using the above-stated method. The peak area of analytes at various concentrations are given in Table 1. The retention time for vitamin D was approximately 4.8 min. The plot of peak area vs concentration of vitamin D3 is shown in Figure 3, the straight line represent that the developed method is linear and there is direct relationship between peak area and concentration. 

The range of this analytical method lies between 80 and 120% of the test concentration. Normally, the acceptance criteria for an analytical method includes the correlation coefficient being ≥0.997. Hence, it was observed that the linearity and range of the current method were in the acceptable bounds, i.e., 0.999. This was calculated using regression line analysis. The representative chromatograms for the linearity and range are shown in Figure 4. 

#### 2.2.3. Accuracy and Recovery

In the study of accuracy and recovery, several samples corresponding to two concentration levels, i.e., 50%, and 100%, were taken and injected using the above-stated method. The results are presented in Table 2. The results indicate that at 50% and 100% concentration the average peak recovery was 49.29% and 99.42%, respectively, while the relative standard deviation was 0.929 and 0.1446%, respectively.

The results were obtained using two concentration levels (50% and 100%) and three replicates of each concentration, with 50% being the lowest concentration and 100% being the highest concentration of the expected working range. The assessment of accuracy was established by evaluating the percentage recovery across the assay range. The acceptance criteria included theoretical amounts ±2%. For 50%, the RSD was 0.929, and for 100%, the RSD was 0.1446; therefore, this method met the evaluation criterion for accuracy. The representative chromatogram for accuracy and recovery is provided below (Figure 5). 

### 2.3. Precision

The precision of the method was checked via the repeatability and ruggedness of the injected samples.

#### 2.3.1. Repeatability (System Precision)

The peak areas of the six replicates of the sample were determined by injecting samples using the above-stated method. The results of precision are shown in Appendix A. The peak areas of the six replicates of the homogeneous sample were determined. The mean value, standard deviation, and relative standard deviation were calculated. The relative standard deviation of six determinations at 100% of the test concentration should not be greater than 2% for the drug product. The RSD of the repeatability precision of the developed method was 0.1741, which lies in the acceptance criteria. The precision repeatability of cholecalciferol (Vitamin D_3_) is provided in Figure 6A. The retention time of vitamin D_3_ is approximately 4.8 min.

#### 2.3.2. Intermediate Precision (Ruggedness)

The intermediate precision, or ruggedness, demonstrates the variation within the lab across several hours and days, while considering various analysts.

##### Within Day’s Variation

The acceptable criteria for intermediate precision (ruggedness) includes an RSD < 2.0%, and the relative standard deviation found for day 1 was 0.059, day 2 was 0.223, and day 3 was 0.396. Hence, the relative standard deviations were found to be in the acceptable range. The chromatogram showing within-day variation is shown in Figure 6B. The within-day variation of the precision ruggedness for the developed method is shown in Table 3. Three samples were checked for three days continuously, standard deviation and relative standard deviation were calculated for these runs as given in Table 3. The values of %RSD were 0.059, 0.223 and 0.396 for day 1, day 2 and day 3 respectively. The lower %RSD values show ruggedness of the developed method. 

##### By Different Analyst 

The acceptance criteria for the precision ruggedness for different analyst variations was RSD < 2.0%, while the resulting RSD was 0.086, which lies in the acceptance criteria. Therefore, the method used is highly rugged. The representative chromatogram is provided in Appendix A, and the numerical data are shown in Appendix A.

#### 2.3.3. Robustness

The robustness parameter was determined during the development of the analytical methods. In the case of liquid chromatography, the mobile phase composition, the role of the column, the flow rate, and the environmental conditions were applicable. For the robustness of the method, this was checked via injecting the samples (*n* = 3) using three different flow rates of 2.0 mL/min, 2.3 mL/min, and 2.5 mL/min, and the RSD calculated was 0.81%. Furthermore, samples were injected using different mobile phase concentrations; the n-hexane/ethyl acetate was used with ratios of 80:20, 85:15, and 90:10, and the RSD was 0.72%. The RSD acceptable criteria for the precision robustness is <3.0%, and it was concluded from the results that the method was truly accepted, as the RSD for the flow rate was 0.81%, while, for the mobile phase composition, it was 0.72%, which lies in the acceptable criteria. The representative chromatograms of the method precision (robustness) flow rate and the mobile phase are given in Appendix A, respectively. 

### 2.4. Limit of Detection (LoD)

The LOD was accomplished at a signal/noise ratio (S/N) of 3. The LOD was experimentally verified with three injections of each vitamin D_3_ at concentrations from 80% to 120% LOD. The calculated LOD was 0.0539 µg/mL, as shown in Appendix A. The LOD is the minimum concentration of an analyte in a sample that can easily be detected, but that is not quantified necessarily. The limit of detection is determined by multiplying the standard deviation with 3.3 and dividing the result by the slope of the curve. The standard deviation of the y-intercepts of the regression lines was used as the standard deviation. 

### 2.5. Limit of Quantification (LoQ) 

The quantification limit was established at a signal/noise ratio (S/N) of 9. The LOQ was experimentally verified using three injections of each vitamin D_3_ at concentrations from 80% to 120% LOD. The calculated LOQ was 0.1633 µg/mL, as shown in Appendix A.

This is the minimum concentration of an analyte in a sample that can be accurately quantified with precision. The LOQ is determined by multiplying the standard deviation by 10 and dividing the result by the slope of the curve. The standard deviation used was the standard deviation of the y-intercepts of the regression line. The determined LOD and LOQ values were 0.0539 µg/mL and 0.1633 µg/mL, respectively, as shown in Appendix A. 

### 2.6. Specificity

The specificity was determined by spiking the sample with the placebo of excipients mixture and the diluent or mobile phase. The results show that the method was not affected by the presence of excipients and the diluent or mobile phase within the range of the peaks of the study. The results of placebo interference and blank interference are shown in Appendix A, respectively. 

### 2.7. Acceptance Criteria

The method should be specific, and the effect of the excipient should be negligible. The excipients did not show significant absorbance at λ max 292 nm. This confirms that there is no interference from excipients at appropriate λ max 292 nm levels of vitamin D_3_. Thus, the method was not affected by the excipient’s presence. The representative chromatogram is provided in Appendix A. 

### 2.8. Application on Commercial Batches

The developed method was applied to the commercially available batches of Osso-D tablets for the further verification of the developed method. For this purpose, the batches selected were 097, 098, and 099, the peak area of the samples and standard are shown in Table 4. 

The developed method was applied to the three commercial batches, i.e., Batch #97, Batch #98, and Batch #99 of Osso-D tablets from Amson Vaccines and Pharma Islamabad, and the results were found to be 100.78%, 102.81%, and 100.81%, respectively as shown in Table 4. From this, it could perhaps be concluded that the method is suitable for tablet forms, and it can be easily applicable to the qualitative and quantitative analysis of vitamin D_3_ via the use of HPLC.

### 2.9. Comparison of the Developed Method with Other Similar Methods

The HPLC method developed in the current study for the quantitative determination of vitamin D3 in tablet form was compared with other HPLC-based methods developed for the analysis of vitamins reported in the literature [36,37,38,39,40]. The comparisons of the various parameters, such as column, mobile phases, elution conditions, sample type, limit of detection and limit of quantification, and the relative standard deviation, etc., of the current method and other similar methods reported in the literature for vitamin D analysis are presented in Table 5. The results show that the coefficient of determination of the current study (0.999) is similar to the previous studies, as shown in Table 5. The percentage recovery of the current method is 99.42%, which is not much more than the previous methods for vitamin D analysis, as shown in Table 5. The limit of detection and the limit of quantification of the current method is close to other methods reported in the literature. These results indicate that a simple norma phase HPLC method was developed for vitamin D3 analysis in pharmaceutical dosage form, which is comparable to the RP-UHPLC and RP-HPLC methods reported in the literature. In this method, a simple silica column was used instead of the expansive C18 column, drawing advantages from the current method over the previously reported methods for vitamin D analysis. 

## 3. Materials and Methods

### 3.1. Selection of Dosage Form

Osso-D tablets from Amson Vaccine and Pharma Islamabad, Pakistan were selected for the study of method development and validation for vitamin D_3_ analysis. Osso-D tablets contain 0.01 mg/tablet vitamin D_3_, while the unit tablet weight was 1030 mg. 

### 3.2. Standard and Sample Preparation

Vitamin D_3_ (7.5 mg) was added to a 100 mL volumetric flask containing 50 mL n-hexane, and was sonicated for 10 min. Then, 2.5 mL of the solution was placed in a 50 mL flask (volumetric), where the volume was adjusted using the same diluent (n-hexane), and this was finally filtered using syringe filters with a pore size of 0.46 µm. The weight of 20 tablets was measured, and these were crushed to a fine powder. The quantity of active ingredients in the tablets was calculated using Equation (1) [41] as follows: (1)$$Quantity\;of\;active\;ingredient\;in\;powder\left(mg\right)=\frac{A\times B}{C}$$
where “A” is weight of the powder obtained from the tablets (weight of 20 tablets in the current study), “B” is the dosage strength mentioned on the medication label (written on the tablet box, such as 10 mg or 20 mg or 500 mg, etc.), and C is the concentration of the active ingredient per tablet (mg). 

### 3.3. HPLC Analysis 

The samples were analyzed using a chromatographic technique, i.e., Shimadzu LC 20AT, equipped with UV detector SPD-20A, Column oven CTO 20A, Pump 20 AD, and controller CBM 20 A. The analysis was carried out using the isocratic elution mode of Shimadzu 20AT HPLC, with the following chromatographic conditions: the mobile phase used was n-Hexane/ethyl acetate (85:15) (*v*/*v*), a detection wavelength of 292 nm, a flow rate of 2.5 mL/min, the column dimension was 4.6 mm × 250cm, the stationary phase was L3, 5 µm (Silica column), and the injection volume was 20 µL.

### 3.4. System Suitability Test

The sample and standard were filtered through a 0.22 μm syringe filter and injected into the standard preparation five times within the chromatograph. The five resulting chromatograms obtained with the injection of the standard preparation were used to calculate the system suitability parameters. The sample (20 μL) was injected into the HPLC system and checked twice. The HPLC analysis of the standard preparation was carried out five times, and the peak areas were measured. The relative standard deviation (%RSD) for the five replicates was measured as less than 2.0%. The results were deduced using the peak area of the chromatograms obtained from standard and sample solutions as follows:(2)$$Assay\left(\%\right)=\frac{Peak\;area\;of\;the\;sample}{Peak\;area\;of\;standard}$$

### 3.5. Method Validation 

#### 3.5.1. System Suitability

The sample was injected six times into the chromatogram, and the peak areas of the six replicates of the sample were determined using this parameter. The standard deviation (SD) and relative standard deviation (% RSD) were calculated using an Excel sheet.

#### 3.5.2. Linearity

To study the linearity concentration ranges of the test substance, the stock solution was prepared using 0.0075 mg/mL of vitamin D3, considered 100%, and then diluted to attain 80% and 90% of that stock solution. The stock solution was also further concentrated to attain 110 and 120% of that stock solution. The samples were injected in triplicate to achieve the reproducibility of the peak areas. Finally, the average of all the peak areas of the above concentrations was calculated, and the coefficient correlation of all the concentrations was calculated [42]. 

#### 3.5.3. Accuracy and Recovery

In the study of accuracy and recovery, a known number of samples corresponding to three concentration levels, i.e., 50%, 100%, and 150%, was prepared. The above-stated concentration, i.e., 0.0075 mg/mL, of the standard was considered 100%. The samples of all three concentrations were injected in triplicate, and the average peak areas were calculated. The assessment of accuracy was established by evaluating the percentage recovery of the analytes across the assay ranges [1].

The percent recovery was calculated using Equation (3) as follows:(3)$$\%Recovery=\frac{Af\times Vf\times Injc\times 100}{Ac\times Vc\times Injf\times n}$$
where Af and Vf are the peak area and volume of the collected fraction; Injc is the injection volume; Ac is the peak area of crude solution; Vc is the crude volume; Injf is the injection volume of the fraction; and n is the dilution factor of the crude sample solution for injection. 

#### 3.5.4. Precision

The six injections of samples were given, and a peak area of the six replicates was determined from the chromatogram. The mean value, standard deviation, and relative standard deviation were determined. The relative standard deviation of the six determinations at 100% of the test concentration should not be greater than 2% for the drug product [43].

#### 3.5.5. Intermediate Precision or Ruggedness 

To calculate the intermediate precision or ruggedness, laboratory variation parameters, such as intra-day variation and different analyst variation, were selected. For intra-day variation, different days were selected, and the samples were injected in triplicates in order to observe day-to-day variations within the results of the proposed analytical method. On the other hand, for the different analyst variation, two analysts were selected to prepare samples, and then HPLC was used to check the variation among the results of the different analysts.

#### 3.5.6. Robustness

The robustness parameter was determined during the development of the analytical method. To check the robustness of the method, the method was checked under different flow rates and different mobile phase compositions. For the change in flow rate, the samples were injected in triplicate at flow rates of 2.4 mL/min, 2.5 mL/min, and 2.6 mL/min. For changes in the mobile phase, the HPLC system was saturated with different mobile phase concentrations, and the samples were injected in triplicates. The selected compositions of the mobile phase were n-hexane/ethyl acetate with the ratios of 83:17 (*v*/*v*), 85:17 (*v*/*v*), and 87:13 (*v*/*v*) [42]. 

### 3.6. Limit of Detection (LoD)

The limit of detection is the minimum concentration of an analyte in a sample that can be detected. The LOD was determined by multiplying the standard deviation by the factor 3.3, and then dividing this by the slope of the curve. By creating a linear regression curve with comparatively low quantities of the target molecule, the limit of quantification was ascertained. The limit of detection (LOD) was calculated from the regression lines, using Equation (4) as follows:(4)$$Limit\;of\;detection\left(LoD\right)=\frac{\left(3.3\times \sigma \right)}{S}$$
where “*σ*” is the standard deviation and (*s*) is the average slope of the calibration curve. 

### 3.7. Limit of Quantification (LoQ)

This is the minimum analyte concentration in any sample that can be accuratly determined. The LOQ was determined by multiplying the standard deviation by factor 10, and then dividing this by the slope of the curve. The LOQ was calculated using Equation (5):(5)$$Limit\;of\;quantification\left(LoQ\right)=\frac{\left(10\times \sigma \right)}{S}$$
where “*σ*” is the standard deviation and (*s*) is the average slope of the calibration curve. 

### 3.8. Specificity

The specificity was determined by spiking the sample with the placebo of the excipients mixture and the diluent or mobile phase. All excipients, diluents, and mobile phases were injected separately in order to check any interference at a specified wavelength, i.e., 292 nm. 

### 3.9. Statistical Analysis

The analyses were repeated three time, and the results were expressed as mean, standard deviation, and relative standard deviation, and the regression line, i.e., correlation coefficient, was calculated. The LOQ and LOD were calculated from the regression lines. 

## 4. Conclusions

A normal phase isocratic HPLC method was developed for vitamin D_3_ analysis with great accuracy, recovery, suitability, precision, and accuracy. The developed method is reproducible, and the repeatability of the method significantly fulfills the official requirements from international protocols. The linearity range of the method is 0.999%, which is an obvious achievement for this new method. This method could be used for the detection and quantification of vitamin D_3_ (Cholecalciferol) in various samples, particularly in tablet form. Moreover, the recovery of the developed method is feasible and achievable. The developed normal-phase HPLC method is simple and robust in comparison to other reported methods; this is because a simple silica column was used for vitamin D analysis instead of an expansive C18 column. The developed method will be utilized for the analysis of other vitamins in tablet forms in the future. 

## Figures and Tables

**Figure 1 pharmaceuticals-17-00505-f001:**
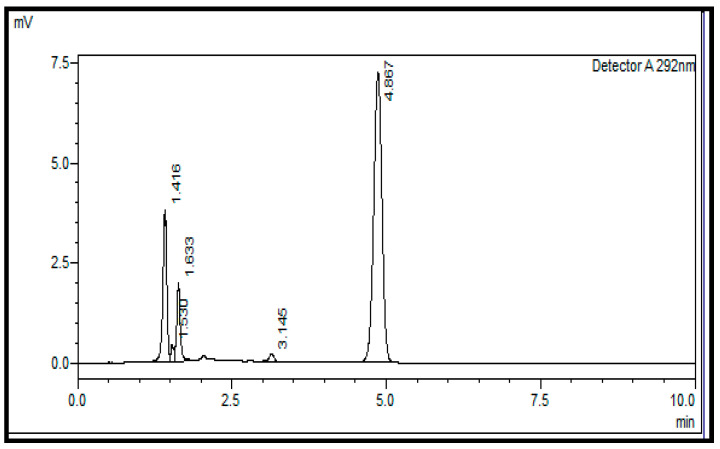
Separation of vitamin D3 (Rt = 4.867 min) using a normal-phase silica column (L3); the mobile phase of n-hexane/ethyl acetate with a ratio of 85:15 (*v*:*v*) and a UV detector wavelength of 292 nm.

**Figure 2 pharmaceuticals-17-00505-f002:**
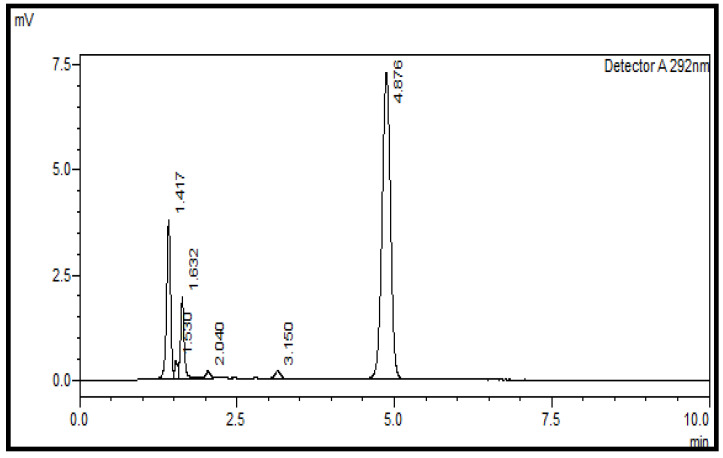
System suitability test for vitamin D_3_ (Rt = 4.867 min) using a normal-phase silica column (L3); mobile phase of n-hexane/ethyl acetate with a ratio of 85:15 (*v*:*v*) and an absorption of 292 nm.

**Figure 3 pharmaceuticals-17-00505-f003:**
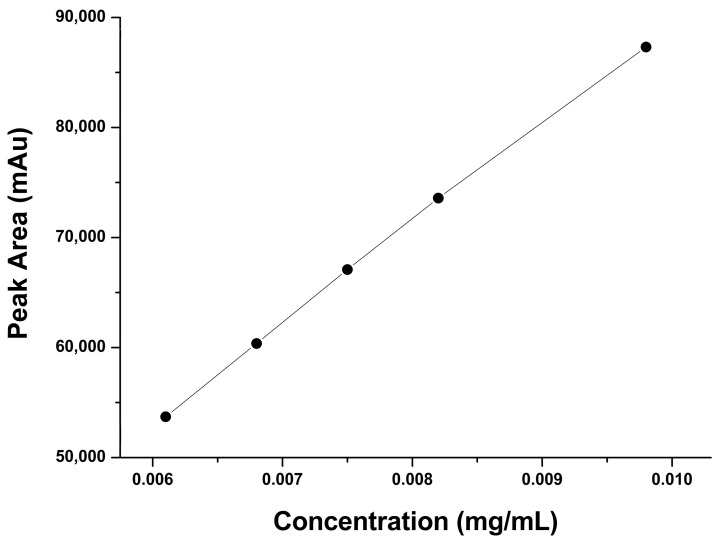
Linearity of the developed method for cholecalciferol (Vitamin D_3_).

**Figure 4 pharmaceuticals-17-00505-f004:**
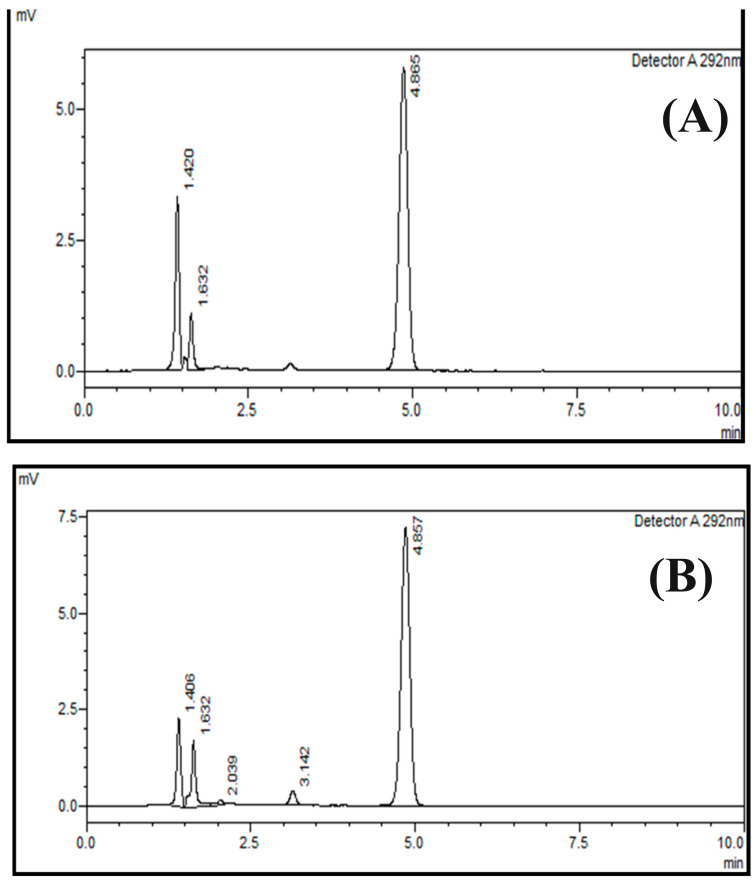
Linearity and range of the HPLC method for cholecalciferol (vitamin D_3_) at 50% (**A**) and 120% (**B**) using a normal-phase silica column (L3); mobile phase of n-hexane/ethyl acetate at a ratio of 85:15 (*v*:*v*) and an absorption of 292 nm.

**Figure 5 pharmaceuticals-17-00505-f005:**
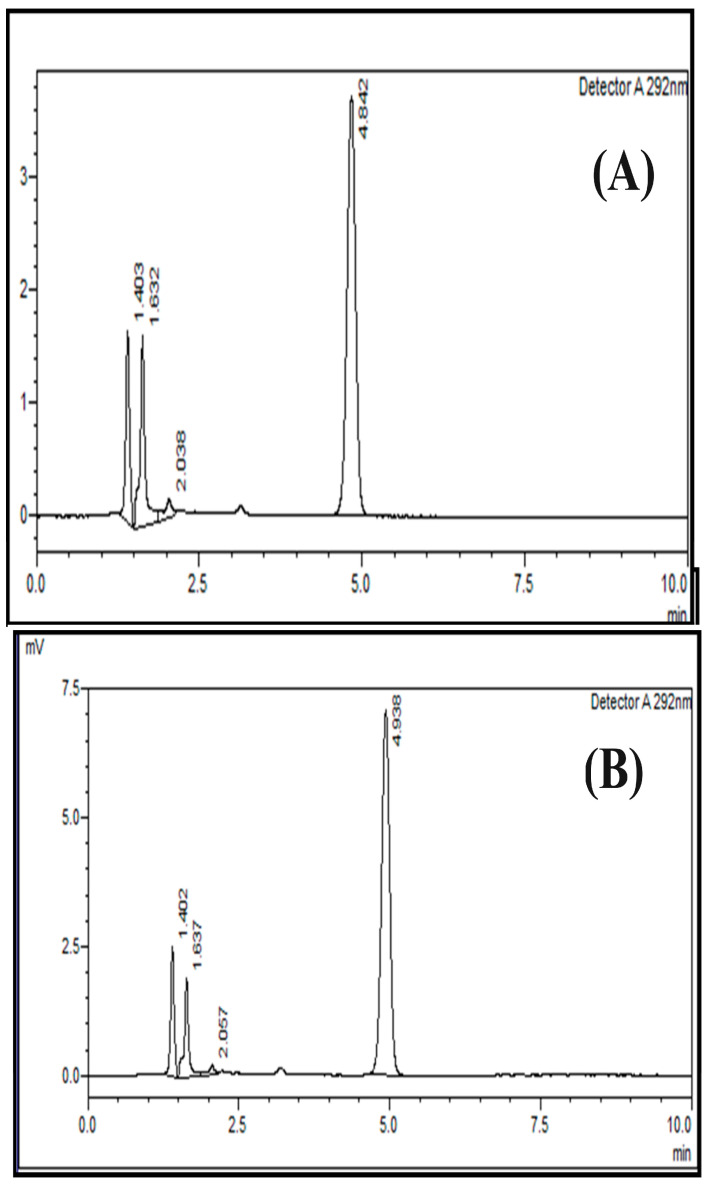
Accuracy and recovery of the HPLC method for cholecalciferol (Vitamin D_3_) at 50% (**A**) and 100% (**B**) using a normal-phase silica column (L3); mobile phase of n-hexane/ethyl acetate at a ratio of 85:15 (*v*:*v*) and an absorption of 292 nm.

**Figure 6 pharmaceuticals-17-00505-f006:**
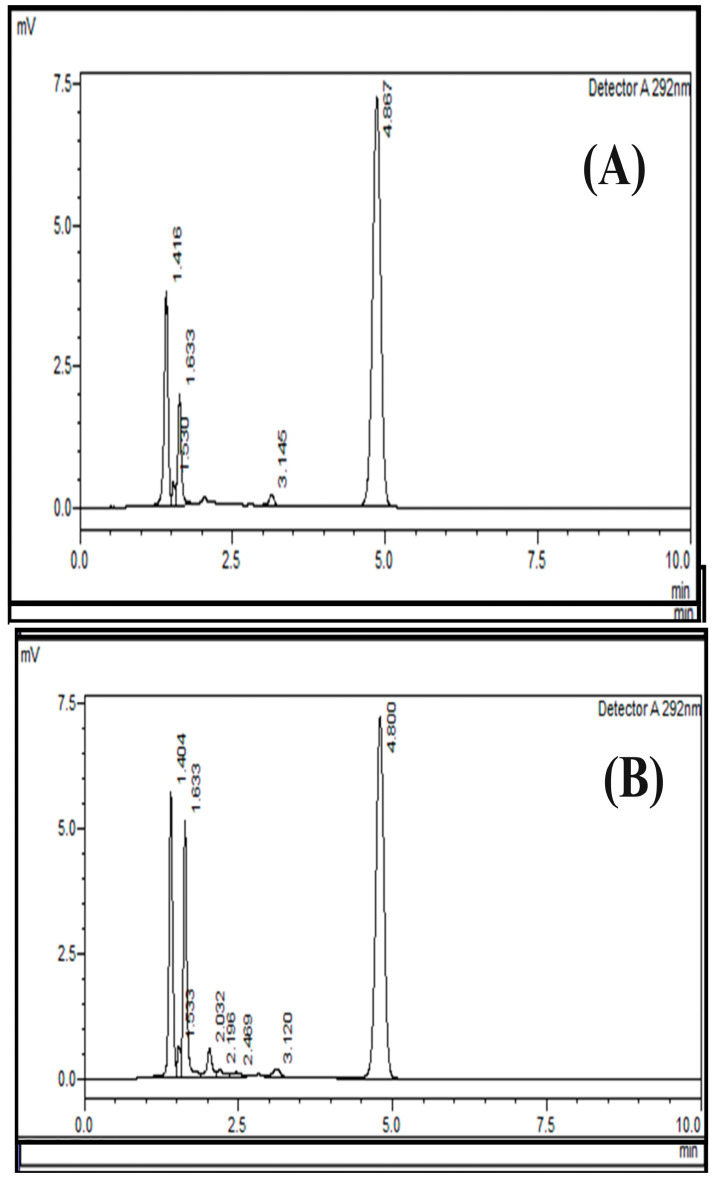
Precision (repeatability) (**A**) and intermediate precision (ruggedness) of the within-day variation (**B**) of the HPLC method for cholecalciferol (vitamin D_3_) analysis using a normal-phase silica column (L3); mobile phase of n-hexane/ethyl acetate at a ratio of 85:15 (*v*:*v*) and an absorption of 292 nm.

**Table 1 pharmaceuticals-17-00505-t001:** Linearity and range of the HPLC method developed for cholecalciferol (Vitamin D_3_) analysis.

Concentration %	Conc. mg/mL	Mean Peak Area
80	0.0061	53,707
90	0.0068	60,367
100	0.0075	67,078
110	0.0082	73,582
120	0.0098	80,315

**Table 2 pharmaceuticals-17-00505-t002:** Accuracy and recovery of the developed method for 50% and 100% cholecalciferol (Vitamin D_3_).

	Conc. mg/mL	Peak Area	%RSD	Recovery (%)	Average Recovery (%)
50%	0.00375	33,869	0.929	50.33	49.92%
		33,257		49.42	
		33,669		50.03	
100%	0.0075	66,838	0.1446	99.31	99.42%
		66,875		99.37	
		67,021		99.58	

**Table 3 pharmaceuticals-17-00505-t003:** Within-day variation of the precision ruggedness for the developed method.

Days	Sample	Assay (%)	Average Assay (%)	STDV	% RSD
1	1	99.26	99.33	0.059	0.059
	2	99.37			
	3	99.35			
2	1	98.68	98.92	0.221	0.223
	2	99.11			
	3	98.97			
3	1	98.91	98.54	0.390	0.396
	2	98.13			
	3	98.57			

**Table 4 pharmaceuticals-17-00505-t004:** Application of the developed method on commercial batches of vitamin D_3_ tablets.

B#	Sample	Peak Area of Sample			Av. Abs. of Sample	Peak Area of Standard			Av. Peak Area of Standard	Assay %	Average Assay %
		1	2	3		1	2	3			
097	1	66,710	66,582	66,476	66,589	66,205	65,661	66,353	66,073	100.78	100.78%
098	1	67,543	67,852	67,509	67,635	65,465	65,691	66,205	65,787	102.81	102.81%
099	1	67,544	67,887	67,358	67,635	66,874	66,587	66,048	65,787	100.81	100.81%

**Table 5 pharmaceuticals-17-00505-t005:** Comparison of the current method with the previously developed HPLC/UHPLC methods for the determination of vitamin D.

Method Developed	Sample/Matrix	Coefficient of Determination (R^2^)	Recovery (%)	Limit of Detection	Limit of Quantification	Ref.
RP-UHPLC-PDA	Pharmaceutical	0.998	98.97	0.24 μg mL^−1^	0.72 μg mL^−1^	[36]
RP-HPLC–UV	Nutritional supplements	0.999	98.3	0.019 mg/L	0.057 mg/L	[37]
RP-UHPLC-UV	Dietary supplements	0.999	82	0.04 µg/mL	0.05 µg/mL	[40]
NP-HPLC-UV	Pharmaceutical	0.999	99.42	0.0539 µg/mL	0.1633 µg/mL	Current study

## Data Availability

The data will be provided by the corresponding author upon written request.

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
