# Peer review of "Development and Validation of Novel HPLC Methods for Quantitative Determination of Vitamin D3 in Tablet Dosage Form"

_pharmaceuticals, 2024, doi:10.3390/ph17040505_

Round 1

Reviewer 1 Report

Comments and Suggestions for Authors

Major comments:

1.       Page 3, lines 113-116. Based on the description here, the concentration of the stock solution is 0.015 mg/ml, but the authors kept saying the stock solution is 0.0075 mg/ml throughout the manuscript (line 140, line 148, etc.)  Section 2.2 needs to be revised if it’s issue with the description.

2.      Page 3, lines 117-119. “The amount of powder was calculated by using the following formula given in Eq (1)”. What is the purpose of this Equation?  This is the amount of powder for what? It is suggested to add more explanation here.

3.      Page 3, lines 139-140. How do you “dilute” something to make it more concentrated than 100%?

4.      Page 5, section 3.1 Method Development. More explanation and discussion are needed in this section to illustrate why this method is selected. What makes it better compared to other methods tried? It is suggested to add chromatograms of other methods to give a visual comparison at least in supplement information.

5.      Page 7-9, section 3.2.2. In section 3.2.2 it says the range lies in 80-120%, then in Figure 4, it says 50-100%. What is the actual linearity range you propose?

6.      Page 10, Table 2. Wrong values were entered for Recovery% and Average recovery% of 100% concentration. Also, what peak area is used as the true 50% and 100% peak area for recovery% calculation? Using peak areas from Table 1 do not give the recovery% the authors wrote in Table 2.

7.      Figure 4 and Figure 5. These 2 figures appear to be useless; they’re not showing anything related to linearity and range, and accuracy and recovery.

8.      Figure 6 and Figure S1. These figures are not showing anything related to repeatability and ruggedness. To be able to show repeatability and ruggedness, chromatogram overlays of repeats need to be provided instead of only showing one chromatogram.

9.      Figure 7. What is the purpose of should these two chromatograms?

Minor comments:

1.      Page 3, line110-111. Extra spaces

2.      Page 4, line 154. “The mean value, relative standard, and deviation Standard deviation” should be “The mean value, standard deviation, and relative Standard deviation”.

3.      Page 5, line 198. “LQ and LD” should be “LOQ and LOD”.

4.      Page 6, line 228. “0.1741” should be “0.1741%”.

5.      Page 10, line 266. For 50% the RSD is not 1.84.

6.      Page 10, lines 264-266. This sentence needs to be rephrased.

7.      Page 11, line 297. “relative standard were calculated” should be “relative standard deviation were calculated”.

8.      Page 14, line 306. “flow rates were 2.0 ml/min” should be “flow rates 2.0 ml/min …”

9.      Page 14, line 313. “Fig. 7” should be “Fig.7 and Fig.8”.

10.   Page 15, line 321. LOD is 0.0539, not 0.0535. Should be Table S5, not S3.

11.   Page 16, line 329. Should be Table S6. Line 339 should be Table S7 and S8.

Comments on the Quality of English Language

Some sentences need to be rephrased or revised, examples are given in "Comments and Suggestions" part.

Author Response

A point by point response to the reviewers comments are given in the file attached below 

Reviewer 2 Report

Comments and Suggestions for Authors

Manuscript “Development and validation of novel HPLC method for quanti-2 tative determination of vitamin D3 in tablet dosage form” deals with the development and validation of the HPLC method for the analysis of vitamin D3. The manuscript describes all the most important parameters of method validation; however, the section comparing the developed method to existing similar methods is missing from the discussion. For example, how this method is improvement in comparison with these two methods: doi: 10.1093/chromsci/bmw048  and 10.1155/2017/1208753

Why were only tablets from one manufacturer evaluated in the study, while other types of tablets were not?

What types of fillers are used in D3 tablets?

Do other fillers interfere with vitamin D3?

What if the tablet contains a combination of D2 and D3 vitamins?

I ask the authors not to repeat the same information in the tables and graphs, and to write the figures' titles so that it is clear what they show.

Can authors explain how they calculated the recovery? Since the recovery was less than 50%, what about the potential degradation products? Can they interfere with the results?

Author Response

(The authors gave the same response as above.)

Reviewer 3 Report

Comments and Suggestions for Authors

Manuscript ID pharmaceuticals-2916312 authored by Dr. Muhammad Saqib Gohar is an interesting study aiming to develop and validate an NP-HPLC method for the quantitative determination of vitamin D3 in tablets. Despite the fact that the manuscript is well written, and many validation parameters were assessed (according to current international guidelines). I am not sure about the novelty. I was able to trace many articles describing vitamin D3 quantification in pharmaceuticals by means of NP and RP HPLC. In my opinion the novelty should be underlined in more details. The current method should be compared with previously published papers in terms of analytical performance criteria and maybe  in terms of greenness (if applicable).

Author Response

(The authors gave the same response as above.)

Reviewer 4 Report

Comments and Suggestions for Authors

The present work concerns the development of a HPLC method for determining vitamin D3 in tablets.

In terms of editing, the work has a number of shortcomings, such as multiple repetitions, typographical errors. Also, the numerical data presented in Table 2 are incorrect.

The work also contains too many chromatograms that do not provide any new information. To me. Some chromatograms from Supplementary Materials are more intersting.

The work looks like a laboratory report on the development of a routine analytical procedure. There is also no proper comparison with other similar works in the literature. There is only general information in this area. In summary, the aspect of scientific novelty is not significant

Also, Introduction section seems to be rather chaotic, e.g.:

„This protein is required to combat different pathogens like mycobacterium tuberculosis in human cells (macrophages) [22]. Therefore, the analysis of vitamin D critically requires a precise and robust method [23].”

Comments on the Quality of English Language

In terms of editing, the work has a number of shortcomings, such as multiple repetitions, typographical errors.

English must be improved.

Author Response

(The authors gave the same response as above.)

Reviewer 5 Report

Comments and Suggestions for Authors

The determination of vitamin D in dietary supplements presented in this paper is not groundbreaking. Vitamin extraction from supplements using hexane in this version is known and used. What is new is the use of a normal phase system for chromatographic separation. This approach eliminates the evaporation of hexane from the extracts.
The work from the point of view of product quality control is interesting and I recommend it for publication. However, I have a few questions and comments:
1. All units for numbers should be written with a space
2. It would be good to add a description in the chromatogram drawings which peak is from vitamin D3. This information is in the text, but the notation in the drawing makes it easier to interpret.
3. What is the LOD and LOD value: quote: "Line 333: The determined LOQ and LOD values were 0.0539 and 0.1633, respectively." The unit is also missing. The first value is LOQ?

Author Response

(The authors gave the same response as above.)

Round 2

Reviewer 2 Report

Comments and Suggestions for Authors

The authors made an effort to improve their manuscript. I suggest just one more small correction of figure 3 - mark the units on the y-axis and align the numbers on the x-axis. After this small correction, the paper can be published.

Author Response

Reviewer’s 2 comments

Reviewer's comment: The authors made an effort to improve their manuscript. I suggest just one more small correction of figure 3 - mark the units on the y-axis and align the numbers on the x-axis. After this small correction, the paper can be published.

Authors response to reviewer’s comments: Thank you so much for appreciating our efforts to improve the quality of our manuscript. The suggested corrections in Fig 3 have been carried out accordingly.

Reviewer 3 Report

Comments and Suggestions for Authors

Dear authors you tried to address all my major concerns. However I am still not convinced about the novelty. Isocratic elution is not enough novelty.  Analytical performance of  not compared with the one of previously published methods. A narrative comparison was performed but not in terms of systematic analysis of analytical performance. Greenness assessment would have been interesting.

Author Response

Reviewer’s 3 comments

Dear authors you tried to address all my major concerns. However I am still not convinced about the novelty. Isocratic elution is not enough novelty.  Analytical performance of not compared with the one of previously published methods. A narrative comparison was performed but not in terms of systematic analysis of analytical performance. Greenness assessment would have been interesting.

Authors response to reviewer’s comments: Dear reviewer, Thank you for appreciating our efforts., we have tried out best to improve the manuscript according to the reviewer recommendations and suggestions.  In the second stage revision, the analytical parameters of the current method are compared with other similar studies reported in the literature and a table (Table 5) as well as text has been added in the revised manuscript from line # 295-315, in which various parameters of method development are compared. We hope this section of the manuscript will be according to your expectations now. 

Reviewer 4 Report

Comments and Suggestions for Authors

The manuscript was corrected according to some Reviewer comments. It sounds better as far as scientific value is concerned. However, some sentences still seem trivial, e.g.”

„Gradient separation is not possible by simple HPLC system which can only be used for isocratic separation.”

Reviewer’s comment 4. Also, Introduction section seems to be rather chaotic, e.g.:„This protein is required to combat different pathogens like mycobacterium tuberculosis in human cells (macrophages) [22]. Therefore, the analysis of vitamin D critically requires aprecise and robust method [23].

I secure that read the entire Introduction. And I further maintain that some sentences are introduced unnecessarily. Every drug requires an appropriately precise and robust method, not only a drug such as vitamin D3.

Comments on the Quality of English Language

 Moderate editing of English language required.

Author Response

Reviewer’s 4 comments

The manuscript was corrected according to some Reviewer comments. It sounds better as far as scientific value is concerned. However, some sentences still seem trivial, e.g.”

„Gradient separation is not possible by simple HPLC system which can only be used for isocratic separation.”

Reviewer’s comment 4. Also, Introduction section seems to be rather chaotic, e.g.:„This protein is required to combat different pathogens like mycobacterium tuberculosis in human cells (macrophages) [22]. Therefore, the analysis of vitamin D critically requires aprecise and robust method [23].

I secure that read the entire Introduction. And I further maintain that some sentences are introduced unnecessarily. Every drug requires an appropriately precise and robust method, not only a drug such as vitamin D3.

Authors response to reviewer comments: Thanks for your valuable comments to improve the quality of our manuscript. This section was carefully reviewed again and appropriate explanation has been provided in the revised manuscript. The introduction was read carefully, and the unnecessary sentences were either deleted or rearranged.

Round 3

Reviewer 3 Report

Comments and Suggestions for Authors

Dear authors you addressed my major concerns. The manuscript became fair enough for publication.